# Immunomodulatory Effects of Multivitamin Complexes Containing *Agaricus blazei* in Patients Undergoing Curative Resection for Non-Small-Cell Lung Cancer: A Randomized, Double-Blind, Placebo-Controlled Multicenter Pilot Trial

**DOI:** 10.3390/biomedicines14010053

**Published:** 2025-12-26

**Authors:** Jiwon Kim, Wonjun Ji, Hyeong Ryul Kim, Geun Dong Lee, Seung Hyeun Lee

**Affiliations:** 1Division of Pulmonary, Allergy, and Critical Care Medicine, Department of Internal Medicine, Kyung Hee University Hospital, College of Medicine, Kyung Hee University, Seoul 02447, Republic of Korea; hi.jiwonk@gmail.com; 2Department of Pulmonary and Critical Care Medicine, Asan Medical Center, College of Medicine, University of Ulsan, Seoul 05505, Republic of Korea; jack1097@naver.com; 3Department of Thoracic and Cardiovascular Surgery, Asan Medical Center, College of Medicine, University of Ulsan, Seoul 05505, Republic of Korea; drhrkim10@gmail.com (H.R.K.); geundonglee010304@gmail.com (G.D.L.); 4Department of Precision Medicine, Graduate School, Kyung Hee University, Seoul 02447, Republic of Korea

**Keywords:** lung cancer, surgery, immunomodulation, *Agaricus blazei*, multivitamin

## Abstract

**Background/Objectives**: Surgical resection of non-small-cell lung cancer (NSCLC) often results in temporary suppression of natural killer cell activity (NKA), potentially increasing the risk of recurrence. This study aimed to evaluate whether multivitamin and mineral complexes containing *Agaricus blazei* could support postoperative immune recovery. **Methods**: In this randomized, double-blind, placebo-controlled multicenter pilot trial, 66 patients with stage I–III NSCLC received either a supplement or a placebo for 28 days post-surgery. NKA was assessed using an interferon-γ release assay preoperatively, on postoperative days (POD) 1–4, and on POD 30. Immune cell subsets, cytokine levels, clinical parameters, and quality of life were evaluated. **Results**: Both groups showed a postoperative decline in NKA, with recovery observed by POD 30. Although the increase in NKA was not statistically significant, the treatment group showed a greater relative recovery (17.8% vs. 9.9%, *p* = 0.104). Immune profiling demonstrated significantly better preservation of T cells (*p* = 0.026) and B cells (*p* = 0.001) as well as a greater reduction in monocytes (*p* = 0.031) in the treatment group. No significant differences were observed in cytokine levels, body mass index, Eastern Cooperative Oncology Group Performance Status, or patient-reported outcomes. **Conclusions**: Supplementation with a multivitamin and mineral complex containing *Agaricus blazei* may contribute to favorable immune modulation in patients undergoing curative surgery for NSCLC. Larger long-term trials are warranted to confirm these findings and facilitate clinical application.

## 1. Introduction

Lung cancer remains the leading cause of cancer-related mortality worldwide, with non-small-cell lung cancer (NSCLC) accounting for approximately 85% of all cases [1]. Despite advances in treatment, the 5-year survival rate of NSCLC remains low, primarily due to late-stage diagnosis and a high rate of recurrence. Surgical resection remains the primary treatment modality for NSCLC. However, postoperative complications, particularly immune suppression and systemic inflammation, can significantly affect patient outcomes and long-term survival [2].

Among the immune system components involved in tumor surveillance, natural killer (NK) cells play a pivotal role in detecting and eliminating malignant cells without prior sensitization [3]. Understanding the postoperative dynamics of NK cell activity (NKA) is essential for improving the prognostic outcomes in patients with NSCLC [4,5]. NK cells, which are innate lymphocytes that exert cytotoxic effects on tumor cells through a coordinated balance between activation and inhibitory signals, recognize target cells via an array of surface receptors that regulate their cytotoxic functions. Tumor cells often evade immune detection by downregulating major histocompatibility complex class I molecules, thereby becoming more susceptible to NK cell-mediated killing [6,7]. In lung cancer, NK cells suppress tumor growth by secreting interferon-gamma (IFN-γ), granzyme B, and perforin. However, this antitumor activity is frequently inhibited by the tumor microenvironment, which releases cytokines, such as interleukin-10 (IL-10) and transforming growth factor-beta (TGF-β), that impair NK cell function [8,9]. In addition, chronic inflammation associated with lung cancer further disrupts NKA, thereby fostering an immunosuppressive milieu that facilitates tumor progression and metastasis.

Surgical resection, particularly in patients with cancer, is accompanied by transient immunosuppression, which may influence postoperative outcomes [10]. From a mechanistic perspective, postoperative immunosuppression is mediated by a complex interplay of neuroendocrine, inflammatory, and immune regulatory pathways. Surgical stress activates the hypothalamic–pituitary–adrenal axis and sympathetic nervous system, leading to increased release of cortisol, catecholamines, and prostaglandins, all of which directly impair NK cell cytotoxicity and T-cell function [11,12]. In parallel, tissue injury induces a surge of immunosuppressive cytokines, including TGF-β and IL-10, which inhibit NK cell activation, promote regulatory T-cell expansion, and suppress antigen presentation [9,13,14]. Moreover, surgery-associated myeloid-derived suppressor cells and monocytes contribute to transient immune paralysis through arginase activity, reactive oxygen species generation, and inhibitory immune checkpoint signaling [14,15]. Collectively, these molecular and cellular mechanisms create a postoperative “immunosuppressive window,” during which antitumor immune surveillance is compromised, potentially facilitating tumor recurrence and metastasis.

Among the immune components, NKA declines markedly within the first 24 h post-surgery and remains suppressed for several days [10]. For example, studies on colorectal and hepatobiliary cancers have shown that over 90% of patients experience profound reductions in NKA immediately after surgery, regardless of clinical stage or surgical extent [16,17]. Notably, NK cell dysfunction or reduced NKA during the acute postoperative period has been linked to a higher risk of recurrence and metastasis [13]. Although partial recovery from NK cell cytotoxicity occurs over time, NKA often remains subnormal until approximately 4–5 weeks postoperatively, with substantial interindividual variability [18]. These findings suggest that perioperative interventions aimed at preserving or accelerating NK cell recovery may be clinically relevant. Therefore, monitoring NKA dynamics provides an objective index of immune recovery after curative surgery, supporting the rationale for immunonutritional or immunomodulatory approaches in postoperative settings.

Given the pivotal role of NK cells in preventing tumor recurrence and their marked suppression following surgery, enhancing postoperative NKA has become a key area of interest. Micronutrients, including essential vitamins and minerals, have been shown to modulate immune function in patients with cancer, particularly by supporting NKA and mitigating postoperative immune suppression [19,20]. Vitamins A, C, D, and E, along with trace elements such as zinc and selenium, contribute to the enhancement of innate immunity and may have synergistic effects when combined with other immunoactive compounds. Specifically, vitamin D supplementation has been shown to improve survival in early-stage lung adenocarcinoma [21], while selenium enhances NKA and cytotoxic lymphocyte expansion [22]. Despite the ongoing research on immunosupportive therapies, targeted strategies to restore NK cell function after NSCLC surgery remain limited and represent a critical unmet clinical need. To address this gap, various interventions, including the use of immunoactive nutraceuticals, have been explored. Among these, *Agaricus blazei*, a medicinal mushroom traditionally used as a dietary supplement, has demonstrated the potential to enhance NKA in patients with cancer [23,24].

Its bioactive constituents, including glucans and polysaccharides, are believed to stimulate NK cell proliferation and boost cytotoxicity [25,26]. Due to its superior β-glucan and ergosterol levels, *A. blazei* offers more potent NKA stimulation and anti-metastatic activity than other medicinal mushrooms [27,28]. Mechanistically, β-glucans from A. blazei activate macrophages and NK cells by binding to Dectin-1 and complement receptor 3 (CR3) [26,29]. *A. blazei* supplementation enhances immune responses and increases NK cell-mediated cytotoxicity, supporting its potential as an adjunct therapy for postoperative immune recovery [29]. In addition, its anti-inflammatory properties may counteract the immunosuppressive effects frequently observed postoperatively, thereby improving patient recovery.

*A. blazei* can enhance NKA and support immune responses; however, its specific effects on restoring NK cell function following surgical resection of NSCLC have not been adequately investigated [13]. This gap in the literature necessitates targeted research to evaluate its therapeutic potential in postoperative settings. Although the immunomodulatory properties of *A. blazei* have been reported in various cancers, its role in facilitating immune recovery after lung cancer surgery remains unclear. While lung cancer-specific clinical data are limited, our rationale is extrapolated from the proven NKA-enhancing effects of *A. blazei* in other malignancies, including gynecological cancers and leukemia, alongside preclinical evidence in lung cancer models [17,19,20]. Addressing this gap is essential in improving patient outcomes and reducing the risk of recurrence by enhancing postoperative immune function. Furthermore, elucidating the mechanisms of action in the context of surgical recovery will provide novel insights into immunotherapeutic strategies for lung cancer.

Postoperative immune suppression, particularly the transient decline in NKA, is a critical yet underexplored determinant of recurrence risk in patients undergoing curative resection for NSCLC. Despite the growing recognition of perioperative immunonutrition, no previous randomized controlled trials have prospectively evaluated the effects of multivitamin- and *A. blazei*-based supplements on postoperative immune recovery in this population. This study aimed to evaluate the effects of multivitamins and mineral complexes, specifically emphasizing the role of *A. blazei* on NKA following NSCLC surgery. The findings of this study provide novel insights into the potential of dietary supplementation as a supportive strategy for restoring immune function after surgical resection, addressing an unmet clinical need.

## 2. Materials and Methods

### 2.1. Study Design

This prospective, randomized, double-blind, placebo-controlled, multicenter trial was conducted at two tertiary referral hospitals, Asan Medical Center and Kyung Hee University Hospital in Seoul, Republic of Korea, between August 2020 and April 2022, with the aim of evaluating immune recovery following curative surgical resection of NSCLC. The participants were randomized in a 1:1 ratio to receive either standard postoperative care (control group) or standard care plus multivitamin-and *A. blazei*-containing supplements (treatment group) for 30 days post-surgery. Randomization was stratified by age and sex to ensure balance between the groups. The study was conducted in accordance with the principles of the Declaration of Helsinki and approved by the Institutional Review Boards of Asan Medical Center (No. 2020-0698) and Kyung Hee University Hospital (No. 2020-01-014). Written informed consent was obtained from all participants before enrollment. The study protocol was registered with the Clinical Research Information Service of the Republic of Korea (KCT0005100).

### 2.2. Study Population

Eligible participants were adults (age ≥ 18 years) diagnosed with NSCLC and scheduled to undergo curative-intent lobectomy. The preoperative diagnosis of NSCLC was confirmed based on definitive histopathological examination by dedicated thoracic pathologists to ensure eligibility. The exclusion criteria included prior chemotherapy or immunotherapy within the past 6 months, known autoimmune disease, immunodeficiency, chronic systemic steroid use, or concurrent participation in another clinical trial. Baseline demographics, medical histories, and preoperative laboratory values were collected before randomization.

The surgical procedures and perioperative management followed institutional standards. Detailed inclusion and exclusion criteria are provided in Appendix A.

A prospective sample size estimation was conducted for the primary endpoint (NK cell activity), assuming an intergroup comparison and an independent two-sample *t*-test. With a two-sided α = 0.05 and power = 0.80, a standardized mean difference of approximately 0.69 (Cohen’s d) was considered clinically meaningful based on prior clinical experience and expected variability. Under these assumptions, 33 patients per group (*n* = 66) would provide 80% power to detect such a difference. Accounting for a 20% attrition rate, the final target enrollment ensured adequate statistical power.

### 2.3. Treatment

Participants were randomly assigned in a 1:1 ratio to either the treatment or control group. Patients in the treatment group received a multivitamin and mineral complex containing *A. blazei* powder (NK365^®^, NKmax Co., Ltd., Seongnam, Republic of Korea; Appendix A), administered twice daily for 28 days, starting between postoperative day (POD) 1 and POD 4. The control group received a matching placebo following the same schedule. The participants were instructed to take four capsules each after breakfast and dinner, for a total of eight capsules per day.

The dosage used in this study (four capsules every 12 h, totaling eight capsules per day) was determined based on prior clinical data evaluating the immunomodulatory and safety profiles of the key active ingredients, β-glucan and *A. blazei* extract. Specifically, β-glucan exerts immune-enhancing effects at doses of 100–400 mg/day in cancer and healthy populations [30,31]. Higher doses (up to approximately 500 mg/day) have also been safely used to reduce fatigue [32]. Similarly, clinical trials of *A. blazei* have confirmed its safety and tolerability across daily intakes of 1.8–5.4 g of mushroom powder [33] and 900 mg of extract with immunomodulatory benefits [34]. In our study formulation, a total daily intake of 2400 mg of *A. blazei* powder and 300 mg of β-glucan was achieved by administering eight capsules per day (four capsules twice daily). Therefore, this dosage was selected to align with previously validated ranges shown to be both effective and well tolerated, ensuring optimal immunologic stimulation within established clinical safety standards.

The placebo formulation was identical to the active supplement in appearance, capsule composition, and excipients, except that the active ingredients (multivitamins, minerals, and *A. blazei* powder) were replaced with inert fillers. Only participants who consumed > 80% of their capsules during the treatment period were included in the final per-protocol (PP) analysis.

### 2.4. Study Endpoints

The primary endpoint was the change in NKA on POD 30 between the treatment and control groups. Secondary endpoints included changes in peripheral immune cell subsets and serum cytokine concentrations (IL-6, IL-10, TGF-β, and TNF-α) over the study period. The key immunologic objective of this trial was to compare POD 30 values and the preoperative baseline to determine whether NKA and immune subsets had recovered to baseline levels 1 month post-surgery. To assess whether differences in immune recovery influenced clinical outcomes, we evaluated and compared clinical parameters between groups, including hospital stay, postoperative complications, quality of life, and recurrence-free survival (RFS). Quality of life was assessed at each visit using the European Organization for Research and Treatment of Cancer Quality of Life Questionnaire–Lung Cancer Module (EORTC QLQ-LC29), which consists of multidimensional scales and single-item questions.

This module is a validated lung cancer-specific module developed to complement the EORTC QLQ-C30 and to evaluate symptoms and functional aspects relevant to patients with lung cancer [27]. The scores of this module range from 0 to 100, with higher scores indicating better functioning or a lower symptom burden, corresponding to improved quality of life. Adverse events were systematically recorded throughout the study to evaluate the safety and tolerability of the interventions.

### 2.5. Sample Collection and Processing

Randomization was performed before the first blood sampling to ensure that baseline preoperative (V1) immune parameters were assessed after group allocation but before any intervention. Angka et al. reported that NKA was suppressed within the first few days following surgery and remained impaired for several weeks thereafter [14]. Accordingly, we selected POD 1–4 (V2) to capture the acute phase of surgery-induced immunosuppression when NKA and cytokine responses are expected to reach their nadir and POD 30 (V3) to evaluate mid-term immune recovery, representing the clinically relevant period when immune reconstitution typically stabilizes after curative resection. At each visit (V1, V2, and V3), 10 mL of peripheral blood was collected in heparinized tubes (BD Biosciences, Franklin Lakes, NJ, USA), of which 1 mL was used to assess the NKA at all three time points. For immune cell subset analysis, 5 mL of blood was collected at V1 and V3 and processed to isolate peripheral blood mononuclear cells (PBMCs). The remaining volume was centrifuged to obtain serum, which was stored at −80 °C until cytokine analysis.

### 2.6. NKA Measurement

NKA was measured using the NK Vue^®^ Kit (NKmax, Seongnam-si, Republic of Korea) according to the manufacturer’s instructions. The NK Vue^®^ assay is a specialized interferon-gamma (IFN-γ) release assay. Unlike conventional cytotoxicity assays, it utilizes Promoca^®^, a proprietary cytokine that selectively stimulates NK cells in whole blood. The concentration of IFN-γ measured in the supernatant by enzyme-linked immunosorbent assay (ELISA) directly reflects the functional status and cytotoxic potential of the circulating NK cells [35]. Briefly, 1 mL of whole blood was placed into an NK Vue^®^ Tube containing Promoca^®^, a proprietary cytokine that selectively stimulates NK cells. After incubation at 37 °C for 20–24 h, the tube was centrifuged at 11,500× *g* for 1 min to collect the supernatant. IFN-γ levels in the supernatant were quantified using an ELISA to determine NKA.

### 2.7. Immune Cell Phenotyping via Flow Cytometry

PBMCs isolated from 5 mL of blood were suspended at a concentration of 5 × 10^6^ to 1 × 10^7^ cells/mL in cryopreservation medium and stored in liquid nitrogen until analysis. For phenotyping, the thawed PBMCs were washed and stained with fluorescently labeled monoclonal antibodies targeting the following surface markers: CD3 (PE, Cat. No. 555333), CD4 (PerCP-Cy 5.5, Cat. No. 560650), CD8 (Alexa Fluor 488, Cat. No. 557696), CD14 (APC, Cat. No. 340436), CD20 (PerCP-Cy 5.5, Cat. No. 340955), CD25 (APC, Cat. No. 555434), and CD56 (FITC, Cat. No. 562794). All antibodies used in this study were purchased from BD Biosciences (Franklin Lakes, NJ, USA). After staining, the cells were washed and analyzed using a FACSLyric flow cytometer (BD Biosciences, Franklin Lakes, NJ, USA). The data were processed using FACSuite™ v1.5 software.

### 2.8. Cytokine Quantification by ELISA

Serum cytokine levels, including IL-6, IL-10, tumor growth factor (TGF)-β, and tumor necrosis factor-alpha (TNF-α), were measured using Quantikine^®^ ELISA kits (R&D Systems, Minneapolis, MN, USA). Blood samples were centrifuged at 1000× *g* for 10 min, and the serum was stored at −80 °C until batch analysis. The following reagents were used: Human IL-6 Quantikine ELISA Kit (Cat. No. D6050, R&D Systems, Minneapolis, MN, USA), Human IL-10 Quantikine ELISA Kit (Cat. No. D1000B, R&D Systems, Minneapolis, MN, USA), Human TGF-β1 Quantikine ELISA Kit (Cat. No. DB100B, R&D Systems, Minneapolis, MN, USA), and Human TNF-α Quantikine ELISA Kit (Cat. No. DTA00D, R&D Systems, Minneapolis, MN, USA).

### 2.9. Statistical Analysis

Continuous variables were compared using either the independent *t*-test or the Wilcoxon rank-sum test, as appropriate. Categorical variables were analyzed using the chi-square or Fisher’s exact tests. Intergroup comparisons at each time point were performed using the independent two-sample *t*-test, whereas within-group changes over time were analyzed using repeated-measures ANOVA or generalized estimating equations (GEEs), as appropriate. The normality of continuous variables was assessed using the Shapiro–Wilk test, and the equality of variances was verified using Levene’s test. When the parametric analysis assumptions were violated, the Wilcoxon rank-sum test was used. RFS, defined as the time from surgery to either recurrence or death, was estimated using the Kaplan–Meier method, and differences between groups were evaluated using the log-rank test. A *p*-value of < 0.05 was considered statistically significant. All statistical analyses were conducted using SPSS software (version 20.0; IBM Corp., Armonk, NY, USA).

## 3. Results

### 3.1. Patient Characteristics

A CONSORT flow diagram illustrating participant enrollment and allocation is shown in Figure 1. Eighty patients were screened for eligibility during the study period, of which 14 (17.5%) were excluded: 11 declined to participate, one did not meet the eligibility criteria, and two were excluded for other reasons. The remaining 66 patients were randomized in a 1:1 ratio into either the treatment (*n* = 32) or control group (*n* = 34). The baseline characteristics of the PP cohort are summarized in Appendix A. The median age was 64 years (range, 34–84 years), and no significant differences were observed in age, sex, smoking history, body mass index, or Eastern Cooperative Oncology Group Performance Status score. The histologic subtype was predominantly adenocarcinoma in both groups (81.3% vs. 88.2%; *p* = 0.597). The pathological stages and types of surgery were well balanced. The baseline quality of life, assessed using the EORTC QLQ-LC29 score, was also comparable between the groups.

A total of 29 patients in the treatment group and 24 in the control group were included in the final analysis. The baseline demographic and clinical characteristics of the PP cohort are presented in Table 1. No significant inter-group differences were observed. Adenocarcinoma was the predominant histological subtype, observed in 82.8% of patients in the treatment group and 95.8% in the control group. Lobectomy was the most commonly performed surgical procedure in both groups (79.3% and 83.3%, respectively).

### 3.2. Dynamics of NKA Before and After Surgery

The changes in NKA levels during the treatment period are summarized in Table 2. Baseline NKA (V1) was 622.2 ± 493.7 pg/mL in the treatment group and 834.4 ± 567.3 pg/mL in the control group, with no significant difference between the groups. All patients exhibited a marked reduction in NKA postoperatively at V2, followed by partial recovery at V3 (from 87.1 ± 65.4 to 979.3 ± 558.3 pg/mL in the treatment group; from 175.7 ± 151.8 to 1028.9 ± 781.6 pg/mL in the control group). NKA levels at V2 and V3 did not significantly differ between the groups. The dynamic changes in NKA across the three time points (V1, V2, and V3) are visually summarized in Figure 2. To further assess NKA recovery, the relative change from V2 to V3 was calculated as (V3–V2)/V2. Although the difference was not statistically significant, the recovery rate of NKA in the treatment group was numerically higher (approximately two-fold) than that in the control group (17.8% vs. 9.9%, *p* = 0.104), suggesting a trend toward more rapid NKA recovery in the treatment group (Table 2).

### 3.3. Dynamics of Immune Cell Subsets

PBMC immune subsets were evaluated before and after surgery using flow cytometry. The dynamic changes in the percentages of various cell populations—including NK cells (CD3^−^CD56^+^), NKT cells (CD3^+^CD56^+^), monocytes (CD14^+^), total T cells (CD3^+^CD56^−^), cytotoxic T cells (CD3^+^CD8^+^), helper T cells (CD3^+^CD4^+^), regulatory T cells (CD3^+^CD4^+^CD25^+^), and B cells (CD20^+^), are summarized in Table 3. At baseline, the proportions of all immune cell subsets were comparable between the two groups. To evaluate the effect of the supplement on immune cell recovery, changes from baseline to POD 30 (V3–V1) were analyzed. No significant differences were observed in NK or NKT cell populations between the groups. However, a significantly greater decrease in monocyte proportion was observed in the treatment group than in the control group (−1.8 ± 5.8% vs. 3.4 ± 9.4%, *p* = 0.031). Additionally, the increase in total T cell population was significantly higher in the treatment group than in the control group (1.0 ± 9.8% vs. −5.0 ± 7.9%, *p* = 0.026). The B cell population also showed a significantly smaller decrease in the treatment group than in the control group (−0.1 ± 3.1% vs. −3.4 ± 3.0%, *p* = 0.001). Among T cell subsets, both CD4^+^ (−1.5 ± 6.3% vs. −5.1 ± 5.7%, *p* = 0.045) and CD8^+^ T cells (1.1 ± 5.0% vs. −1.2 ± 2.7%, *p* = 0.043) showed more favorable change at V3 in the treatment group, contributing to the overall rise in total T cell counts.

### 3.4. Dynamics of Different Cytokines

Dynamic changes in cytokine levels, including those of IL-6, IL-10, TNF-α, and TGF-β, are summarized in Table 4. Baseline cytokine levels did not differ significantly between the two groups. Among these cytokines, IL-6 increased markedly from 3.8 ± 5.6 to 54.7 ± 56.5 pg/mL in the treatment group and from 3.2 ± 3.1 to 57.8 ± 55.7 pg/mL in the control group at V2 compared with that at V1. Similarly, IL-10 levels rose from 4.0 ± 3.4 to 13.2 ± 13.4 pg/mL in the treatment group and from 4.3 ± 4.6 to 10.4 ± 5.6 pg/mL in the control group. In both groups, the elevated levels returned to the baseline values by V3. In contrast, TNF-α and TGF-β levels did not show similar postoperative fluctuations. Additionally, no significant differences in the recovery of cytokine levels were observed between the two groups (V3–V1) (Table 4).

### 3.5. Post-Surgical Complications and Quality-of-Life Score

Post-surgical complications and quality-of-life scores are summarized in Table 5. No significant differences were observed between the groups in the mean length of hospital stay (11.6 ± 6.6 days in the treatment group vs. 9.7 ± 6.2 days in the control group, *p* = 0.305) or the incidence of postoperative complications (12.5% vs. 17.6%, *p* = 0.734). In addition, the change in EORTC QLQ-LC29 scores from before to after surgery did not significantly differ between the two groups (11.1 ± 8.7 in the treatment group vs. 9.6 ± 5.4 in the control group, *p* = 0.465). To assess postoperative quality of life (QoL), we further compared the five symptom-related multidimensional items of the module, including pain at the surgical site at V3; however, no significant intergroup differences were identified (Appendix A).

### 3.6. Long-Term Surgical Outcomes

The median follow-up period for the study population was 48.8 months (range, 12.5–57.3 months), during which 12 patients (22.6%) experienced recurrence or death: seven (24.1%) in the treatment group and five (20.8%) in the control group. The Kaplan–Meier curves for RFS in the overall population and each study group are presented in Appendix A, and the 3- and 5-year RFS rates are summarized in Table 6. The median RFS was not achieved in either group. The 3- and 5-year RFS rates in the overall population were 80.8% and 72.7%, respectively. No significant differences in 3- (71.4% in the treatment group vs. 81.7% in the control group, *p* = 0.415) and 5-year RFS rates (65.5% in the treatment group vs. 78.3% in the control group, *p* = 0.377) were observed between the groups. 

### 3.7. Safety

All reported adverse events (AEs) are listed in Table 7. Overall, 24.5% (13/53) of participants experienced AEs. The difference in AE incidence between the two groups was not statistically significant (*p* = 0.305): 37.9% and 20.8% in the treatment and control groups, respectively. Most AEs were gastrointestinal and included anorexia, nausea, and dyspepsia. No significant difference was observed in the incidence of AEs between the two groups. All reported AEs were mild to moderate in severity, with no Grade ≥ 3 AEs observed.

## 4. Discussion

*A. blazei* and essential micronutrients can modulate immune responses in patients with cancer [26,32]. *A. blazei* activates NK cells, macrophages, and T lymphocytes through β-glucan–mediated mechanisms. Clinical studies involving patients with gynecological and gastrointestinal cancers have reported enhanced NK cell cytotoxicity and increased cytokine production following oral *A. blazei* administration [30]. Furthermore, in a randomized trial, Ahn et al. found that daily administration of *A. blazei* extract significantly enhanced NKA in patients undergoing chemotherapy and improved appetite loss and general fatigue, among other symptoms [25]. Similarly, vitamins and minerals play important roles in maintaining immune competence, particularly in patients with cancer. For example, vitamin D regulates gene expression in human PBMCs and is essential for the proper development and function of NK cells through receptor-mediated mechanisms [19]. In addition, folate deficiency impairs the proliferation of primary human CD8+ T cells, potentially compromising cell-mediated immune responses [36]. Se enhances antitumor immunity by activating immune effector cells and reversing tumor-induced immunosuppression [37]. The supplements used in this study contained various immune-supportive vitamins and minerals, including vitamin D, folate, and selenium, which provided a biological rationale for the observed immune-enhancing effects (Appendix A). While previous epidemiological studies have indicated that supplemental vitamins may not be associated with a decreased risk of lung cancer development, our study focuses on the perioperative recovery phase. In this context, micronutrients are utilized not for cancer prevention but as immunomodulatory agents to mitigate surgery-induced immunosuppression.

In the present study, we evaluated the immunomodulatory effects, postoperative complications, and clinical outcomes of *A. blazei* supplementation in patients with NSCLC who underwent surgical resection. Although not statistically significant, a prominent relative increase in NKA (approximately two-fold) was observed post-surgery in the treatment group.

Notably, both T and B cell populations showed significantly more favorable trajectories from baseline in the treatment group, as evidenced by increases in CD4^+^ helper T cells and CD8^+^ cytotoxic T cells. The supplement was well-tolerated with no adverse events clearly attributable to its use. These results suggest that supplementation with multivitamins and mineral complexes containing *A. blazei* may contribute to favorable immune modulation in patients who have undergone curative surgery.

NK cells contribute to antimetastatic defense primarily through IFN-γ secretion, which promotes the recruitment, activation, and cytotoxicity of other immune effector cells [34]. Moreover, NK cells play a key role in eliminating circulating tumor cells, and postoperative dysfunction of these cells has been linked to an increased risk of recurrence and reduced overall survival [31]. NKA is affected by multiple factors, including age, sex, and smoking status [38], and is suppressed in patients with cancer, including lung cancer, and impaired NK cell function is associated with poor prognosis [39,40,41]. Surgical stress is a well-established cause of transient immunosuppression; NK cells are particularly susceptible to stress during the early postoperative period [42]. More importantly, accumulating evidence suggests that transient impairment of NK cell cytotoxicity following surgery has been linked to cancer recurrence and metastasis in both animal models and human studies [13,17]. The acute postoperative period is increasingly recognized as a ‘metastatic window’ of vulnerability, during which surgery-induced suppression of NK cells can facilitate the survival and dissemination of circulating tumor cells [43]. Therefore, monitoring NKA at POD 30 is clinically significant, as the timely recovery of NKA has been identified as a critical prognostic indicator for long-term survival after radical surgery. NKA recovery following surgery appears to be affected by multiple factors, including age and the type of surgery. Data on recovery time from NKA following cancer surgery remain limited. Hong et al. reported that suppressed NKA began to increase 3 weeks post-surgery and returned to normal levels at 5 weeks in patients undergoing biliopancreatic cancer surgery [16]. Similarly, Angka et al. reported that NKA was significantly suppressed on POD 1 and remained suppressed until POD 28 in 65.5% of patients who underwent surgery for colorectal cancer [18].

Given the critical role of NK cells in antitumor immunity and metastasis prevention, strategies to preserve or restore NK cell function during the perioperative period are clinically important. In this study, NKA was measured at V1, V2, and V3 using an IFN-γ release assay. Consistent with previous findings [33,34], the NKA levels declined significantly in the immediate postoperative period and returned to baseline by POD 30 in both groups. Although the between-group differences in NKA recovery were not statistically significant, the treatment group showed a greater relative increase in NKA levels from V2 to V3. To the best of our knowledge, this is the first study to demonstrate dynamic changes in NKA in patients who underwent lung cancer surgery and to evaluate the effect of a supplement on perioperative NKA dynamics. The IFN-γ release assay used in our study is a clinically practical and reproducible method for quantifying NKA, with several advantages over conventional cytotoxicity or degranulation-based assays [18,39,44]. The lack of significant differences in NKA in our study may be attributed to the limited sample size and/or the suboptimal timing of sampling post-surgery. Collectively, the findings of this study emphasize the importance of monitoring NK cell dynamics and support continued investigation of immunonutritional strategies to preserve innate immune function during the postoperative period.

Herein, the recovery of total T and B cell populations on POD 30 was significantly prominent in the treatment group, and the increase in T cells was associated with an increase in CD4^+^ and CD8^+^ T cell populations. Concurrently, the treatment group showed a greater decrease in monocyte count. Innate immune cells, such as neutrophils and monocytes, recover quickly after a surgical insult, whereas the restoration of T and B cell function is typically delayed [36]. Wang et al. reported a similar sequential recovery pattern in liver transplant recipients, observing early innate immune activation, followed by gradual reconstitution of adaptive subsets over several months [45].

Likewise, Lachmann et al. found that innate immune functions, including neutrophil and monocyte activities, recovered within days postoperatively, whereas adaptive immune responses remained suppressed for longer durations [44]. Although this phenotypic expansion does not necessarily indicate superior immune competence, our findings suggest that postoperative immune recovery follows a phased progression, with the early dominance of innate responses gradually giving way to the reconstitution of adaptive immunity. In this process, a more prominent NKA in the treatment may contribute to the relative preservation or rapid restoration of adaptive immunity and immune homeostasis. This hypothesis is supported by evidence showing the key role of NKs as major producers of IFN-γ during the early phase of the innate immune response and that they contribute to the initiation of the adaptive immune response [9]. Future studies that incorporate longitudinal sampling and functional assays are warranted to validate these phenotypic trends and clarify their clinical relevance.

NKA is modulated by a broad network of cytokines, including IL-2, IL-12, IL-15, and IL-18, all of which play critical roles in enhancing NK cell proliferation, survival, and cytotoxicity in response to tumors [4]. Although no significant differences were noted in serum levels of IL-6, IL-10, TNF-α, and TGF-β between groups in this study, these cytokines likely represent only a subset of the regulatory factors influencing NK cell cytotoxicity. While these favorable immunological shifts did not lead to immediate improvements in patient-reported QoL or short-term clinical outcomes, the enhanced preservation of adaptive immune subsets suggests a potential for improved host defense during the vulnerable recovery period. In addition, the changes in immune cell subsets reflect the phased progression of postoperative recovery, where innate responses typically recover quickly, while adaptive immunity (T and B cells) restoration is often delayed. In our study, the treatment group showed significantly better preservation of T and B cells, including CD4+ and CD8+ subsets. Mechanistically, a more prominent NKA in the early phase may contribute to the rapid restoration of adaptive immune homeostasis via early secretion of cytokines like IFN-γ. The absence of significant differences in QoL and clinical outcomes may reflect the limitations of this pilot study, including the relatively brief 28-day intervention and small cohort size, which may be insufficient to detect changes in multidimensional patient-reported outcomes. Future studies may benefit from incorporating additional cytokine profiling during long-term treatment to characterize dynamic postoperative immune responses and their potential impact on surgical outcomes.

This study has some limitations. First, as a pilot trial, the sample size was limited and might have lacked sufficient power to detect small effect sizes across all outcomes. Second, the treatment period was confined to 30 days, which may have been sufficient to capture early innate immune responses but insufficient to observe long-term adaptive immune reconstitution or clinically meaningful outcomes, such as recurrence. Third, although the IFN-γ release assay and PBMC subset profiling provided valuable insights into systemic immune dynamics, they do not fully capture the local immune responses within the tumor microenvironment. Fourth, we did not perform in vitro mechanistic validation studies to investigate the cellular and molecular mechanisms of action of this micronutrient. In vitro studies using lung cancer cell lines and immune cell models would enable a more detailed mechanistic understanding, facilitate hypothesis-driven experimental designs, and potentially contribute to the development of personalized immunomodulatory strategies. Finally, although the complex showed immunomodulatory effects, these did not translate into improved clinical outcomes, such as reduced postoperative complications or recurrence, limiting the generalizability of the findings. However, our study comprehensively evaluated the dynamics of immune recovery and clinical outcomes, including short- and long-term surgical outcomes. The measurable immunological effects of this supplement during the postoperative recovery period highlight its potential as a practical adjunctive strategy to support immune recovery following cancer surgery.

## 5. Conclusions

This pilot study suggests a possible trend toward more favorable immune recovery patterns following postoperative administration of a multivitamin and mineral complex containing *A. blazei* in patients with NSCLC, warranting further validation in larger long-term studies. The observed trends in NKA and preservation of adaptive immune cell populations support the biological plausibility of postoperative immunonutritional modulation. Further studies with larger cohorts and extended follow-up periods, and those that incorporate mechanistic analyses of both systemic and tumor-localized immune responses, are warranted. Ultimately, integrating immune-supportive strategies during the postoperative period may enhance host defenses, reduce the risk of recurrence, and improve long-term outcomes in patients undergoing surgical treatment for cancer.

## Figures and Tables

**Figure 1 biomedicines-14-00053-f001:**
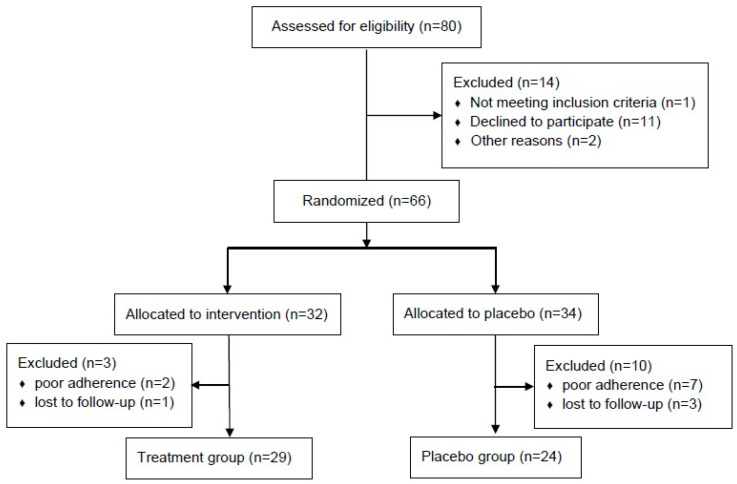
CONSORT flow diagram.

**Figure 2 biomedicines-14-00053-f002:**
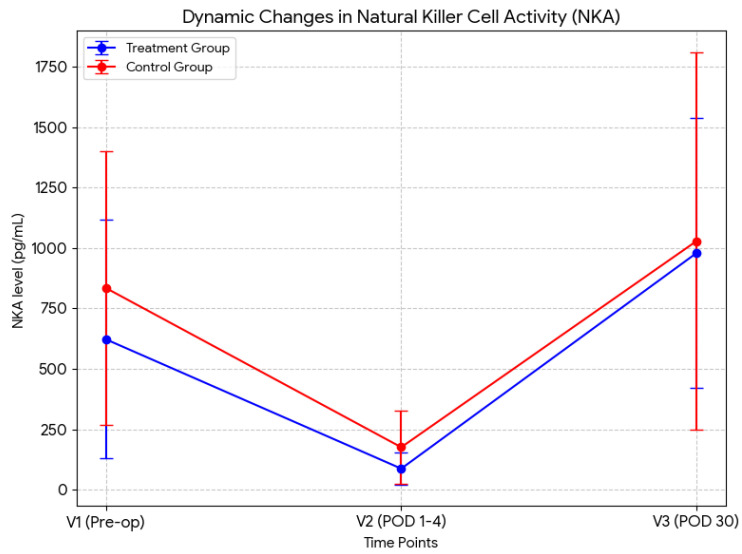
Dynamic changes in NK cell activity (NKA). The graph clearly illustrates the profound suppression of NKA immediately following surgery (V2) compared with baseline (V1), and the subsequent restoration by POD 30 (V3), with the treatment group exhibiting a tendency of steeper recovery slope compared to the control group. POD, postoperative day.

**Table 1 biomedicines-14-00053-t001:** Baseline characteristics of the per-protocol population.

	Overall (*n* = 53)	Treatment Group (*n* = 29)	Control Group (*n* = 24)	*p*-Value
Age (median, range)	64 (34–84)	65 (34–84)	64 (49–73)	0.648
Female sex	22 (41.5)	13 (44.8)	9 (37.5)	0.590
Smoking history				0.710
Never	25 (47.1)	15 (51.7)	10 (41.7)	
Ex-smoker	11 (20.7)	5 (17.2)	6 (41.7)	
Current smoker	17 (32.0)	9 (31.0)	8 (33.3)	
BMI (kg/m^2^)	24.2 ± 2.6	24.1 ± 2.8	24.2 ± 3.1	0.816
Histologic subtype				0.362
Adenocarcinoma	47 (88.7)	24 (82.8)	23 (95.8)	
Squamous cell carcinoma	5 (9.4)	4 (13.8)	1 (4.2)	
Adenosquamous carcinoma	1 (1.8)	1 (3.5)	0	
Pathologic stage				0.648
IA	25 (47.1)	12 (41.4)	13 (54.2)	
IB	11 (20.7)	6 (20.7)	5 (20.8)	
IIB	7 (13.2)	5 (17.2)	2 (8.3)	
IIIA	7 (13.2)	3 (10.3)	4 (16.7)	
IIIB	2 (3.7)	2 (6.9)	0	
IVA	1 (1.8)	1 (3.5)	0	
Type of surgery				>0.999
Lobectomy	43 (81.1)	23 (79.3)	20 (83.3)	
Segmentectomy	6 (11.3)	3 (10.3)	3 (12.5)	
Wedge resection	4 (7.5)	3 (10.3)	1 (4.2)	
Adjuvant chemotherapy *				>0.999
Yes	14 (26.4)	8 (27.5)	6 (25.0)	
No	39 (73.5)	21 (72.5)	18 (75.0)	
Prolonged (≥7 days) post-operative antibiotics				>0.999
Yes	6 (11.3)	3 (10.3)	3 (12.5)	
No	47 (88.7)	26 (89.7)	21 (87.5)	
ECOG PS				0.394
0	21 (39.6)	13 (44.8)	8 (33.3)	
1	32 (60.4)	16 (55.2)	16 (66.7)	
EORTC QLQ-LC29 score	31.8 ± 4.4	32.2 ± 5.0	30.5 ± 4.6	0.189

* Adjuvant chemotherapy was initiated at least 30 days post-surgery according to the guidelines. BMI, body mass index; ECOG PS, Eastern Cooperative Oncology Group Performance Status; EORTC QLQ-LC29, European Organization for Research and Treatment of Cancer Quality of Life Ques-tionnaire–Lung Cancer Module.

**Table 2 biomedicines-14-00053-t002:** Dynamic changes in NKA before and after surgery between the groups.

	Overall (*n* = 53)	Treatment Group (*n* = 29)	Control Group (*n* = 24)	*p*-Value *
NKA, pg/mL				
V1	718.0 ± 536.6	622.2 ± 493.7	834.4 ± 567.3	0.344
V2	125.5 ± 332.9	87.1 ± 65.4	175.7 ± 151.8	0.355
V3	993.0 ± 565.8	979.3 ± 558.3	1028.9 ± 781.6	0.847
Absolute difference in NKA, pg/mL				
V2–V1	−592.5 ± 204.6	−530.8 ± 337.9	−658.7 ± 397.1	0.551
V3–V1	283.2 ± 132.1	376.5 ± 130.4	181.4 ± 117.0	0.284
V3–V2	871.7 ± 550.6	885.1 ± 549.5	838.3 ± 498.7	0.957
Recovery rate				
(V3–V2)/V2	12.6 ± 5.2	17.8 ± 8.3	9.9 ± 8.5	0.104

Data are presented as mean ± standard deviation. * *p*-values were derived using independent *t*-tests. NKA, Natural killer cell activity; V1, preoperative; V2, postoperative days 1–4; V3, postoperative day 30.

**Table 3 biomedicines-14-00053-t003:** Dynamic changes in immune cell subsets.

	Overall (*n* = 53)	Treatment Group (*n* = 29)	Control Group (*n* = 24)	*p*-Value
NK cell, %				
V1	15.6 ± 7.7	14.3 ± 7.8	16.7 ± 7.7	0.348
V3	19.0 ± 9.6	16.3 ± 8.5	19.8 ± 10.0	0.167
V3–V1	3.4 ± 5.8	2.0 ± 4.4	3.1 ± 6.6	0.104
NKT cell, %				
V1	7.2 ± 5.9	8.3 ± 6.5	6.1 ± 5.2	0.204
V3	7.1 ± 7.7	8.2 ± 9.0	6.0 ± 6.0	0.344
V3–V1	−0.1 ± 3.4	−0.1 ± 4.1	−0.1 ± 2.4	0.947
Monocyte, %				
V1	9.8 ± 8.0	9.8 ± 8.5	9.8 ± 7.6	0.978
V3	10.5 ± 10.2	8.0 ± 7.8	13.2 ± 11.7	0.079
V3–V1	0.7 ± 8.1	−1.8 ± 5.8	3.4 ± 9.4	0.031
Total T cell, %				
V1	47.9 ± 10.4	49.3 ± 12.0	46.8 ± 8.6	0.420
V3	46.1 ± 11.4	50.3 ± 10.2	41.8 ± 11.1	0.009
V3–V1	−1.7 ± 9.5	1.0 ± 9.8	−5.0 ± 7.9	0.026
B cell, %				
V1	9.7 ± 4.6	8.6 ± 5.1	10.7 ± 3.7	0.121
V3	8.0 ± 5.1	8.5 ± 7.0	7.3 ± 2.8	0.426
V3–V1	−1.7 ± 3.5	−0.1 ± 3.1	−3.4 ± 3.0	0.001
CD 4^+^ T cell, %				
V1	31.9 ± 9.3	33.9 ± 10.0	29.9 ± 8.1	0.144
V3	28.7 ± 9.7	32.4 ± 9.9	24.8 ± 8.0	0.006
V3–V1	−3.3 ± 6.2	−1.5 ± 6.3	−5.1 ± 5.7	0.045
CD 8^+^ T cell, %				
V1	20.2 ± 6.9	21.5 ± 7.9	18.9 ± 5.6	0.192
V3	20.2 ± 8.3	22.6 ± 9.2	17.7 ± 6.4	0.039
V3–V1	−0.1 ± 4.2	1.1 ± 5.0	−1.2 ± 2.7	0.043
Regulatory T cell, %				
V1	0.8 ± 0.8	0.9 ± 0.9	0.7 ± 0.6	0.500
V3	0.8 ± 0.8	0.9 ± 1.0	0.7 ± 0.5	0.266
V3–V1	0.0 ± 0.4	0.0 ± 0.4	−0.1 ± 0.4	0.368

**Table 4 biomedicines-14-00053-t004:** Dynamic changes in different cytokine levels.

	Overall (*n* = 53)	Treatment Group (*n* = 29)	Control Group (*n* = 24)	*p*-Value
IL-6, pg/mL				
V1	3.5 ± 4.5	3.8 ± 5.6	3.2 ± 3.1	0.648
V2	56.2 ± 55.5	54.7 ± 56.5	57.8 ± 55.7	0.849
V3	3.6 ± 2.6	3.5 ± 2.0	3.8 ± 3.0	0.604
V3–V1	0.1 ± 4.3	−0.4 ± 5.5	0.6 ± 2.5	0.426
IL-10, pg/mL				
V1	4.1 ± 4.0	4.0 ± 3.4	4.3 ± 4.6	0.827
V2	11.8 ± 10.3	13.2 ± 13.4	10.4 ± 5.6	0.355
V3	4.5 ± 4.4	4.9 ± 4.9	4.1 ± 3.8	0.505
V3–V1	0.4 ± 2.7	0.9 ± 2.8	−0.2 ± 2.5	0.157
TNF-α, pg/mL				
V1	2.4 ± 2.7	2.0 ± 2.6	2.9 ± 2.7	0.277
V2	4.2 ± 3.7	3.7 ± 3.6	4.7 ± 3.8	0.353
V3	2.2 ± 2.4	2.4 ± 2.8	2.3 ± 1.8	0.919
V3–V1	−0.3 ± 2.7	0.4 ± 2.8	−0.6 ± 2.7	0.256
TGF-β, pg/mL				
V1	168.5 ± 60.9	165.5 ± 69.7	171.5 ± 51.5	0.741
V2	137.5 ± 49.3	134.9 ± 46.1	145.3 ± 44.1	0.434
V3	171 ± 60.2	164.6 ± 60.5	177.7 ± 60.5	0.464
V3–V1	2.5 ± 53.8	−1.0 ± 50.8	6.1 ± 57.6	0.656

Data are presented as mean ± standard deviation. IL-6, interleukin-6; IL-10, interleukin-10; TNF-α, tumor necrosis factor-α; TGF-β, tumor growth factor-β.

**Table 5 biomedicines-14-00053-t005:** Post-surgical complications and quality of life score.

Characteristics	Overall (*n* = 53)	Treatment Group (*n* = 29)	Control Group (*n* = 24)	*p*-Value
HD, days	10.7 ± 5.8	11.6 ± 6.6	9.7 ± 6.2	0.732
Post-op complications	8 (15.4)	3 (10.7)	5 (20.8)	0.447
Bleeding	0 (0.0)	0 (0.0)	0 (0.0)	>0.999
Pain	1 (1.9)	1 (3.6)	0 (0.0)	>0.999
Pneumonia	3 (5.8)	1 (3.6)	2 (8.3)	0.590
Pneumothorax	2 (3.8)	1 (3.6)	1 (4.2)	>0.999
BPF	0 (0.0)	0 (0.0)	0 (0.0)	>0.999
Lung abscess	0 (0.0)	0 (0.0)	0 (0.0)	>0.999
ARDS	2 (3.8)	0 (0.0)	2 (8.3)	0.208
ICU admission	2 (3.8)	0 (0.0)	2 (8.3)	0.208
Death	0 (0.0)	0 (0.0)	0 (0.0)	>0.999
EORTC QLQ-LC29 score				
V1	31.4 ± 4.0	32.2 ± 5.0	30.5 ± 4.6	0.605
V2	43.6 ± 8.6	44.1 ± 9.1	43.2 ± 9.0	0.303
V3	41.9 ± 7.5	43.4 ± 9.5	40.1 ± 4.9	0.131
V3–V1	10.4 ± 6.0	11.1 ± 8.7	9.6 ± 5.4	0.429

Data are presented as mean ± standard deviation.

**Table 6 biomedicines-14-00053-t006:** Recurrence-free survival.

	Overall (*n* = 53)	Treatment Group (*n* = 29)	Control Group (*n* = 24)	*p*-Value *
Median RFS, month (95% CI)	NR (NA–NA)	NR (49.4–NA)	NR (NA–NA)	0.370
3-year RFS, % (95% CI)	80.8 (70.7–92.2)	71.4 (56.5–90.3)	81.7 (75.3–100.0)	0.415
5-year RFS, % (95% CI)	72.7 (60.7–87.1)	65.5 (49.0–87.5)	76.3 (64.1–100.0)	0.377

* log-rank test *p*-value CI, Confidence interval; NR, not reached; NA, not assessed; RFS, recurrence-free survival.

**Table 7 biomedicines-14-00053-t007:** Safety profiles of the two groups.

Adverse Events	Overall (*n* = 53)	Treatment Group (*n* = 29)		Control Group (*n* = 24)		*p*-Value
		Any Grade	≥Grade 3	Any Grade	≥Grade 3	
All	13 (24.5)	11 (37.9)	0	5 (20.8)	0	0.305
Nausea	4 (7.6)	1 (3.5)	0	3 (12.5)	0	
Anorexia	2 (3.8)	1 (3.5)	0	1 (4.2)	0	
Dyspepsia	1 (1.9)	1 (3.5)	0	0	0	
Constipation	2 (3.8)	2 (6.9)	0	0	0	
Cough	2 (3.8)	1 (3.5)	0	1 (4.2)	0	
Chest pain	1 (1.9)	1 (3.5)	0	0	0	
Myalgia	1 (1.9)	1 (3.5)	0	0	0	
Eye pain	1 (1.9)	1 (3.5)	0	0	0	
Paresthesia	1 (1.9)	1 (3.5)	0	0	0	
Drug eruption	1 (1.9)	1 (3.5)	0	0	0	

Data are presented as numbers (%).

## Data Availability

The original data generated and analyzed in this study are included in the article and its Appendix A. Please contact the corresponding author for further inquiry.

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
