# Peer review of "Immunomodulatory Effects of Multivitamin Complexes Containing Agaricus blazei in Patients Undergoing Curative Resection for Non-Small-Cell Lung Cancer: A Randomized, Double-Blind, Placebo-Controlled Multicenter Pilot Trial"

_biomedicines, 2025, doi:10.3390/biomedicines14010053_

Round 1
Reviewer 1 Report
Comments and Suggestions for Authors
This clinical trial demonstrates the therapeutic benefits of multivital momplexes containing Agaricus blazei for lung cancer patients. The experimental design is reasonable, the results are not excessive, and the discussion and conclusions are appropriate. There is not a single image in the article. If possible, some relevant figures can be provided.
Author Response
We thank you and the reviewers for the comments on our manuscript titled “Immunomodulatory Effects of Multivitamin Complexes Containing Agaricus blazei in Patients Undergoing Curative Resection for Non-Small Cell Lung Cancer: A Randomized, Double-Blind, Placebo-Controlled Multicenter Pilot Trial” (Manuscript ID: biomedicines-4042525).
The manuscript has been revised in response to your thoughtful comments. We have included our point-by-point responses to the reviewers’ comments below. We hope that our manuscript is now suitable for publication in the Biomedicines.
Reviewer #1
This clinical trial demonstrates the therapeutic benefits of multivital complexes containing Agaricus blazei for lung cancer patients. The experimental design is reasonable, the results are not excessive, and the discussion and conclusions are appropriate.
Comment 1: There is not a single image in the article. If possible, some relevant figures can be provided.
Response 1: Thank you for this valuable comment. We agree that visual representation is crucial for clarity. Accordingly, we have added Figure 2 (Line graph of NKA changes from V1 to V3) to illustrate the dynamic immune recovery patterns observed in the study (p.8).
Reviewer 2 Report
Comments and Suggestions for Authors
- Please discuss key molecular mechanisms implicated in the postoperative immunosuppression in the introduction section.
- In the introduction, the authors are failed to provide clinical evidences supporting robust NK-mediated benefits of vitamins and trace elements, particularly in NSCLC.
- The authors should provide a strong rationale for choosing Agaricus blazei over other immunomodulators.
- Supplemental multivitamins, vitamin C, vitamin E, and folate were not associated with a decreased risk of lung cancer.
- The manuscript states that Agaricus blazei enhances NK cell activity in patients with cancer; however, no prior studies specifically demonstrating its effects in NSCLC or other lung cancers are cited. Please clarify whether lung cancer–specific evidence exists or explicitly state that this rationale is extrapolated from studies in other malignancies.
- The manuscript mentions immunomodulatory effects of Agaricus blazei and multivitamins but does not cite specific studies or mechanisms. Brief supporting evidence would strengthen the rationale.
- The clinical significance of the primary and secondary immune endpoints, particularly NKA recovery at POD 30, should be clarified.
- The authors indicate that NKA was measured using the NK Vue® IFN-γ release assay; however, a brief explanation of how ELISA-based IFN-γ measurement reflects NK cell activity would help readers better understand the evaluation strategy. Providing a short description of the assay principle, beyond citing the manufacturer’s protocol, would improve clarity.
- Although the study shows changes in immune parameters, there were no significant differences in postoperative quality of life between the groups. The authors should clarify the clinical relevance of their findings and explain how the observed immunological effects may translate into meaningful benefits for patients.
- In Section 3.6, could the authors clarify how the observed changes in immune cell subsets might be mechanistically relevant, especially since there were no significant differences in clinical outcomes?
- Could the authors, if possible, provide more details on how NSCLC was confirmed in the study participants? Specifically, were histopathological reports or CT scan findings available to validate the diagnosis and ensure eligibility?
- The discussion is quite extensive. Could the authors consider shortening it to focus more on the main findings and their mechanistic relevance, while streamlining parts that are less directly related to the study’s conclusions?
Author Response
We thank you and the reviewers for the comments on our manuscript titled “Immunomodulatory Effects of Multivitamin Complexes Containing Agaricus blazei in Patients Undergoing Curative Resection for Non-Small Cell Lung Cancer: A Randomized, Double-Blind, Placebo-Controlled Multicenter Pilot Trial” (Manuscript ID: biomedicines-4042525).
The manuscript has been revised in response to your thoughtful comments. We have included our point-by-point responses to the reviewers’ comments below. We hope that our manuscript is now suitable for publication in the Biomedicines.
Comment 1: Please discuss key molecular mechanisms implicated in the postoperative immunosuppression in the introduction section.
Response 1: Thank you for this valuable comment. We have now added a dedicated paragraph detailing the key molecular mechanisms underlying postoperative immunosuppression, including neuroendocrine stress responses, immunosuppressive cytokine signaling, and innate immune cell dysfunction (p.2, lines 63-77).
Comment 2: In the introduction, the authors are failed to provide clinical evidences supporting robust NK-mediated benefits of vitamins and trace elements, particularly in NSCLC.
Response 2: Following the reviewer's constructive suggestion, we have revised the Introduction to include clinical evidence specifically demonstrating the benefits of micronutrients in patients with NSCLC. Specifically, we cited a randomized controlled trial showing that Vitamin D supplementation improves survival in early-stage lung adenocarcinoma patients (Akiba et al., 2018). We also included evidence that selenium supplementation enhances NK cell activity and stimulates the expansion of cytotoxic lymphocytes (Kiremidjian-Schumacher et al., 1994). These additions strengthen the clinical rationale for our study within the specific context of lung cancer (p.3, lines 93-100).
Comment 3: The authors should provide a strong rationale for choosing Agaricus blazei over other immunomodulators.
Response 3: We appreciate this important request for clarification. Agaricus blazei was selected as the key immunomodulator in our study due to its superior concentration of beta-glucans compared to other medicinal mushrooms (Firenzouli F. el al., 2008). Furthermore, unlike many other supplements, A. blazei contains specific compounds such as ergosterol, which have been proven to inhibit tumor-induced angiogenesis and metastasis in preclinical models (Takaku T. et al., 2001). We have added these points to the Introduction (p.2, lines 107-111) to provide a stronger rationale for our selection of this specific compound.
Comment 4: Supplemental multivitamins, vitamin C, vitamin E, and folate were not associated with a decreased risk of lung cancer.
Response 4: We appreciate the reviewer’s insightful comment regarding the role of micronutrients in cancer prevention. We fully agree that large-scale epidemiological evidence has not consistently demonstrated a risk-reduction effect of supplemental vitamins on the development of lung cancer. However, it is important to clarify that the conceptual framework of our study is fundamentally distinct from cancer prevention in the general population. The primary focus of this trial is the acute perioperative period, where surgical stress induces transient but profound immunosuppression. In this specific clinical context, these micronutrients are utilized as short-term immunomodulatory agents to support the restoration of host defenses, a goal that is clinically and mechanistically separate from long-term cancer prophylaxis. We have emphasized this distinction in the Discussion section to ensure the focus of our study is clearly understood (p. 12, lines 413-417).
Comment 5: The manuscript states that Agaricus blazei enhances NK cell activity in patients with cancer; however, no prior studies specifically demonstrating its effects in NSCLC or other lung cancers are cited. Please clarify whether lung cancer–specific evidence exists or explicitly state that this rationale is extrapolated from studies in other malignancies.
Response 5: We agree with this comment. While specific clinical data for Agaricus blazei in NSCLC is limited, our rationale is based on its proven NKA-enhancing effects in other cancers including gynecological cancers and leukemia, alongside preclinical evidence in lung cancer models. We have explicitly clarified this extrapolation in the Introduction (p.3, lines 121-124).
Comment 6: The manuscript mentions immunomodulatory effects of Agaricus blazei and multivitamins but does not cite specific studies or mechanisms. Brief supporting evidence would strengthen the rationale.
Response 6: We have strengthened the rationale by adding specific references detailing the immunomodulatory mechanisms of A. blazei. Specifically, we have described how its β-glucans activate innate immunity by binding to Dectin-1 and complement receptor 3 (CR3) on macrophages and NK cells, subsequently enhancing their tumoricidal activity in the Introduction (p.3, lines 107-111).
Comment 7: The clinical significance of the primary and secondary immune endpoints, particularly NKA recovery at POD 30, should be clarified.
Response 7: We appreciate the reviewer’s insightful comment regarding the clinical relevance of our endpoints. Postoperative suppression of NKA, particularly during the early acute phase, is increasingly recognized as a "metastatic window" that can facilitate the survival of circulating tumor cells and lead to recurrence (Tai, L.H. et al, 2013). We selected POD 30 as a primary endpoint because it represents a critical juncture where acute surgical stress and inflammatory responses typically subside, allowing for the assessment of immune homeostasis stabilization. More importantly, recent clinical evidence has identified the timely recovery of NKA as a significant prognostic indicator for long-term survival in patients undergoing radical surgery for cancer. Therefore, monitoring NKA at POD 30 serves as an objective index to evaluate whether a patient has successfully transitioned out of the vulnerable postoperative period. We have clarified this significance in the Discussion section (p. 13, lines 441-446).
Comment 8: The authors indicate that NKA was measured using the NK Vue® IFN-γ release assay; however, a brief explanation of how ELISA-based IFN-γ measurement reflects NK cell activity would help readers better understand the evaluation strategy. Providing a short description of the assay principle, beyond citing the manufacturer’s protocol, would improve clarity.
Response 8: We have added a detailed explanation of the NK Vue® assay principle in Section 2.6, clarifying that it measures IFN-γ release specifically from stimulated NK cells to reflect their functional status (p.5, lines 232-236).
Comment 9: Although the study shows changes in immune parameters, there were no significant differences in postoperative quality of life between the groups. The authors should clarify the clinical relevance of their findings and explain how the observed immunological effects may translate into meaningful benefits for patients.
Response 9: We appreciate the reviewer’s thoughtful observation regarding the quality of life (QoL) results. We agree that the observed immunological improvements in the treatment group did not translate into immediate, measurable benefits in subjective QoL or short-term clinical outcomes. However, the significant preservation of T and B cell populations represents a meaningful biological effect, indicating a robust restoration of adaptive immunity during the critical postoperative period. We recognize the lack of divergence in QoL scores as a limitation of this pilot trial, likely due to the small sample size and the short 28-day intervention, which may be insufficient to detect changes in complex, multidimensional QoL scales. We have revised the Discussion section to address this more transparently, clarifying that these objective immunological effects should be viewed as a precursor to potential long-term clinical benefits (p. 14, line 496-508).
Comment 10: In Section 3.6, could the authors clarify how the observed changes in immune cell subsets might be mechanistically relevant, especially since there were no significant differences in clinical outcomes?
Response 10: The changes in immune cell subsets reflect the phased progression of postoperative recovery, where innate responses typically recover quickly, while adaptive immunity (T and B cells) restoration is often delayed. In our study, the treatment group showed significantly better preservation of T and B cells, including CD4+ and CD8+ subsets. Mechanistically, a more prominent NKA in the early phase may contribute to the rapid restoration of adaptive immune homeostasis via early secretion of cytokines like IFN-γ. We have revised the Discussion section to address this point (p. 14, line 496-508).
Comment 11: Could the authors, if possible, provide more details on how NSCLC was confirmed in the study participants? Specifically, were histopathological reports or CT scan findings available to validate the diagnosis and ensure eligibility?
Response 11: We have clarified in Section 2.2 that the diagnosis was confirmed via preoperative histopathological examination by expert pathologists (p. 3, lines 156-158).
Comment 12: The discussion is quite extensive. Could the authors consider shortening it to focus more on the main findings and their mechanistic relevance, while streamlining parts that are less directly related to the study’s conclusions?
Response 12: We appreciate the reviewer’s thoughtful comment. We have streamlined the Discussion section, focusing more on the mechanistic relevance of our findings while reducing general descriptions of micronutrient.
Reviewer 3 Report
Comments and Suggestions for Authors
Dear Authors,
This manuscript represents a well-designed and carefully conducted pilot study investigating the immunomodulatory potential of a multivitamin-mineral supplement containing Agaricus blazei in patients with non-small cell lung cancer (NSCLC) undergoing curative surgical resection.
The title of the paper is clearly and concisely stated and indicates the structure and topic of the paper, i.e. the objectives of this study are clearly defined, and the conclusions drawn are in accordance with the presented results and in accordance with the title of the paper.
The study is methodologically well-designed and conducted, and the results obtained are in accordance with the applied methodology.
The manuscript is based on scientific material obtained by reviewing the literature and data available to date.
The topic of the paper is clinically relevant, and the study was methodologically well designed and conducted, and the results obtained provide preliminary but important insights into perioperative immunomodulation in patients with non-small cell lung cancer (NSCLC).
The abstract is clear and concise, fully aligned with the study objectives, while the introduction is informative, well-supported by the literature, and clearly explains the rationale for the choice of the research concept. The authors successfully link the clinical problem of NSLC and postoperative immunosuppression to relevant biological mechanisms involving NK cells and propose supplementation with Agaricus blazei as a potential immunonutrient.
The Materials and Methods section is clearly and systematically structured, with sufficient detail to enable full reproducibility of the study. The study design, characteristics of the study population, description of the intervention, defined goals and endpoints, and procedures are presented precisely and clearly. The groups are clearly defined (randomized, double-blind, placebo-controlled multicenter study) which is a significant methodological advantage and contributes to a high level of internal validity, but again, although the statistics are clearly stated and performed, there are not many results that are statistically significant, could this indicate that future studies may need to increase the sample size, i.e. the number of patients, or certain parameters as such simply do not significantly affect the result? The quality and scope of the applied methods are at an exceptionally high level, and the focus areas of the research are clearly highlighted.
The results are clearly presented and interpreted with appropriate caution. The statistical processing of the results is clearly stated, but once again, check the accuracy of all the tables you cite and whether they show results that correlate with the statistics of the processed results themselves and pay attention to the uniform presentation of the tables themselves and the agendas you place next to them.
As a suggestion, a thorough check of statistical significance and appropriate interpretations in all tables, especially Table 3, is recommended to ensure consistency between numerical results and textual description.
The study provides encouraging preliminary evidence that supplementation with a multivitamin-mineral complex containing Agaricus blazei may favorably modulate postoperative immune recovery in patients with NSLC. Although the absence of statistically significant differences in the primary outcome requires cautious interpretation, the observed trends in immune subsets warrant further investigation in larger and adequately powered studies.
The discussion is detailed, well-structured, and clearly links the study findings to the existing literature on the immunomodulatory effects of Agaricus blazei, micronutrients, and perioperative immunosuppression in oncology patients.
However, several limitations should be highlighted, one of which may be the lack of studies investigating the mechanism of action of this micronutrient; prediction on, for example, cell lines of this type of cancer would be important, which would open opportunities for more complex studies and provide a significant contribution to personalized therapy.
The conclusion is clear, coherent and based on the presented results, and the manuscript represents a valuable pilot study and a solid basis for future larger studies.
The references are appropriate and relevant to the topic, and the conclusions are well aligned with the study findings, providing a coherent and accurate summary of the research.
Regards,
Author Response
We thank you and the reviewers for the comments on our manuscript titled “Immunomodulatory Effects of Multivitamin Complexes Containing Agaricus blazei in Patients Undergoing Curative Resection for Non-Small Cell Lung Cancer: A Randomized, Double-Blind, Placebo-Controlled Multicenter Pilot Trial” (Manuscript ID: biomedicines-4042525).
The manuscript has been revised in response to your thoughtful comments. We have included our point-by-point responses to the reviewers’ comments below. We hope that our manuscript is now suitable for publication in the Biomedicines.
Comment 1: The groups are clearly defined (randomized, double-blind, placebo-controlled multicenter study) which is a significant methodological advantage and contributes to a high level of internal validity, but again, although the statistics are clearly stated and performed, there are not many results that are statistically significant, could this indicate that future studies may need to increase the sample size, i.e. the number of patients, or certain parameters as such simply do not significantly affect the result?
Response 1: We acknowledge this as a primary limitation. As this was a pilot study, the sample size may have been insufficient to reach statistical significance for all parameters. This has been emphasized in the Conclusion as a rationale for larger subsequent trials (p.15, line 531-533).
Comment 2: The statistical processing of the results is clearly stated, but once again, check the accuracy of all the tables you cite and whether they show results that correlate with the statistics of the processed results themselves and pay attention to the uniform presentation of the tables themselves and the agendas you place next to them.
Response 2: We appreciate the reviewer’s meticulous observation regarding the accuracy and presentation of our data. We have conducted a comprehensive, point-by-point re-verification of all tables (Tables 1–7) to ensure that every numerical value accurately reflects the results of the processed statistical analyses. Additionally, we have standardized the formatting of all tables, including the consistent placement of legends and abbreviations, to ensure a uniform and professional presentation throughout the manuscript.
Comment 3: As a suggestion, a thorough check of statistical significance and appropriate interpretations in all tables, especially Table 3, is recommended to ensure consistency between numerical results and textual description.
Response 3: Following the reviewer's suggestion, we have performed a thorough audit of the statistical significance and interpretations across all tables, with a particular focus on Table 3. We have ensured that the textual descriptions in the Results (Section 3.3) and Discussion sections are perfectly consistent with the p-values and numerical results provided in the tables. Any minor discrepancies in interpretation have been corrected to ensure that our conclusions are strictly grounded in the presented data.
Comment 4: However, several limitations should be highlighted, one of which may be the lack of studies investigating the mechanism of action of this micronutrient; prediction on, for example, cell lines of this type of cancer would be important, which would open opportunities for more complex studies and provide a significant contribution to personalized therapy.
Response 4: We thank the reviewer for this insightful comment. We have added this point to the Discussion as another limitation, explicitly acknowledging the absence of in vitro mechanistic validation and emphasizing the importance of future cell line–based studies to elucidate the mechanism of action and support personalized therapeutic approaches (p. 15, lines 518-522).
Round 2
Reviewer 2 Report
Comments and Suggestions for Authors
I found that the authors sufficiently revised the manuscript. In addition, the authors have provided a nicely detailed and thorough response to the comments from the previous review and have addressed my concerns. This revision has significantly improved the manuscript. I enjoyed reading it. In my view, this manuscript can be published in the present revised form now.